# Correlation of Hand Grip Strength with Sleep Quality and Perception of General Health Status in University Students: A Cross-Sectional Study

**DOI:** 10.3390/jfmk10020122

**Published:** 2025-04-05

**Authors:** Jaime Ruiz-Tovar, Jorge Mendoza, Mathis Corral, Tim Desgranges, Marcela Marcial, Alexandra Rivilla, Noellia Perez, Angel Sacedo, María Simarro-Gonzalez, Ana Martin-Nieto

**Affiliations:** 1San Juan de Dios Foundation, 28036 Madrid, Spain; msimarro@comillas.edu (M.S.-G.); amartinn@comillas.edu (A.M.-N.); 2Health Sciences Department, San Juan de Dios School of Nursing and Physical Therapy, Comillas Pontifical University, 28036 Madrid, Spain; 202220335@alu.comillas.edu (T.D.); marcelamarcial@alu.comillas.edu (M.M.); 202218721@alu.comillas.edu (A.R.); 202306840@alu.comillas.edu (N.P.); angel.sacedo@alu.comillas.edu (A.S.); 3Physical Therapy Department, San Juan de Dios School of Nursing and Physical Therapy, Antonio de Nebrija University, 28036 Madrid, Spain; jmendozag2@alumnos.nebrija.es (J.M.); mcorral1@alumnos.nebrija.es (M.C.)

**Keywords:** sleep quality, emotional disorders, hand grip strength, physical activity, physical therapy, Pittsburgh sleep quality index questionnaire, GHQ-12, IPAQ, dynamometer

## Abstract

**Background/Objectives:** The aims of this study were to establish the relationship between hand grip strength (HGS) and sleep disturbances, as well as to correlate HGS with the perception of general health status. **Methods:** A cross-sectional study was conducted among Physical Therapy students. Participants completed the International Physical Activity Questionnaire–Short Form (IPAQ-SF), the Pittsburgh Sleep Quality Index (PSQI), and the General Health Questionnaire (GHQ-12). HGS was measured using a dynamometer and self-reported anonymously. **Results:** A total of 145 students participated (58.6% males; mean age: 21.0 ± 3.9 years). The average HGS was 42.4 kg in the dominant hand and 39.2 kg in the non-dominant one. Poor subjective sleep quality was reported by 27.5%; 84.1% slept less than 7 h. GHQ-12 scores indicated that 31.7% may be experiencing emotional distress. HGS was inversely correlated with PSQI scores in both dominant (ρ = –0.211; *p* = 0.019) and non-dominant hands (ρ = –0.178; *p* = 0.049). Students with GHQ-12 scores >12 had significantly lower HGS. No significant correlation was found between HGS and physical activity intensity. **Conclusions**: Lower hand grip strength was correlated with poor sleep quality and higher GHQ-12 scores, independently of physical activity levels. These findings suggest that HGS may serve as a simple and accessible indicator of psychological vulnerability in university students.

## 1. Introduction

Hand grip strength (HGS) is one of the methods classically used to assist in the early identification of physical frailty and sarcopenia, mainly in older adults. Studies have found a relationship between HGS and overall mortality, associated with the presence and rapid progression of various pathologies, including cardiovascular disease, diabetes mellitus, cancer, and hip fractures in older adults [1,2,3]. Although this association has been demonstrated in older adults, a relationship with mortality in middle-aged individuals has also been found in China [4].

Fortunately, the mortality and prevalence of chronic diseases in the young population is very low. Therefore, there are no studies that demonstrate a relationship between physical diseases and HGS. Gomez-Campos et al. have shown that there is a nonlinear relationship between chronological age and HGS from childhood to senescence, with men experiencing an accelerated and continuous reduction in HGS and women experiencing a reduction in HGS from the age of 40 years onwards [5]. This shows that apart from the natural aging process that begins in the third decade of life, there are other factors that influence HGS, such as gender, physical activity, or lifestyle habits [5].

A recently published study among Spanish university students showed that 29% of them perceived their state of health as fair or poor, which is not associated with diagnosis of chronic pathologies, but rather with stress and insufficient sleep. Poor health was also associated with excessive abuse of alcohol and other psychoactive drugs, and a tendency to live a sedentary lifestyle [6].

Sleep disturbances have also been correlated with reduced HGS and sarcopenia in older adults, due to low muscle strength or poor physical performance, showing a direct association with poor quality of life. Sleep plays an important role in the functioning of cells, organs, and systems. Several studies have shown that altered sleep duration is associated with an increased risk of mortality in middle-aged and older individuals. Insufficient sleep can reduce the level of growth hormones (GHs) and increase cortisol level and proinflammatory markers, which in turn elevate the risk of cardiovascular disease, insulin resistance, diabetes, and obesity [7,8]. Several studies have suggested that both short sleep duration and excessive sleep duration (collectively referred to as pathological sleep) are positively associated with sarcopenia [9,10,11,12].

Although there is limited evidence in the literature of the association between sleep disturbance and HGS in young adults, some early associations have been observed in this population. These may serve as potential indicators of future physical and psychological conditions, such as sarcopenia or anxiety–depression disorders, allowing for early intervention to modify the progression of these pathologies [1,2,3].

University students represent a population undergoing significant physical, psychological, and behavioral changes. During this life stage, habits such as sleep patterns, physical activity, nutrition, and substance use are often redefined, which can influence both current and future health outcomes. Assessing HGS in this demographic may provide valuable insights into early indicators of muscular fitness, lifestyle balance, and overall well-being. Furthermore, identifying early declines in physical function or associations with poor sleep and perceived health may help in developing preventive strategies tailored to young adults, potentially reducing future risk of chronic diseases or psychological conditions.

Based on the previous literature, we hypothesize that lower HGS may be correlated with lower sleep quality and perceived poorer general health in young adults. The aims of this study were to examine the relationship between HGS and sleep disturbances and to explore the association between HGS and perceived general health status.

## 2. Materials and Methods

A cross-sectional observational study was conducted. Consecutive non-probability sampling was performed for the recruitment of participants pursuing a Physical Therapy degree at the San Rafael campus of the San Juan de Dios School of Nursing and Physical Therapy, Comillas Pontifical University, during the academic year 2024–2025. A total of 145 participants were included in the study. Exclusion criteria were a personal history of neuromuscular pathology that alters the determination of HGS, current or anatomical lesions within the past three months that hinder hand mobility, and sequelae of previous trauma or surgery or congenital alterations that alter the determination of HGS. Additionally, students who were not proficient in Spanish were excluded, as the questionnaires were only available and validated in Spanish.

### 2.1. Methodology

For data collection, a questionnaire was designed on Google Forms^®^ (Google, Mountain View, CA, USA), compiling questions on sociodemographic and anthropometric variables from the physical activity questionnaire (IPAQ-SF) validated for the Spanish population [6], the Pittsburgh Sleep Quality Index (PSQI) validated for the Spanish population [13], and the General Health Questionnaire (GSQ-12) validated for the Spanish population [14]. The short version of the IPAQ has demonstrated good reliability in university student populations, with Cronbach’s alpha values ranging from 0.65 to 0.88 depending on the domain, and test–retest intraclass correlation coefficients (ICCs) around 0.76 [6]. The PSQI has shown adequate internal consistency in adult populations, with Cronbach’s alpha values typically ranging from 0.70 to 0.83 in various studies [13]. The GHQ-12 is a mental health screening test, containing 12 questions about health in the last few weeks. The GHQ-12 has demonstrated high internal consistency in general Spanish-speaking populations, with Cronbach’s alpha values usually above 0.85 [14]. As part of this questionnaire was on Google Forms^®^, and to ensure the anonymization of the data, a variable was added where the participant would record the HGS quantification established by dynamometer.

The IPAQ-SF Questionnaire consists of 6 questions, referring to the type of physical activity performed for how long and how often. From these data, a calculation of METS (a metabolic rate measurement unit that establishes the intensity of an activity) is achieved, and the physical activity is classified into Sedentary: <80 METS, Mild activity: 81–600 METS, Moderate activity: 601–1500 METS, and Vigorous activity: >1501 METS. Analysis of physical activity intensity was included to assess whether it influenced HGS independently of sleep quality or perceived health status [6].

The PSQI contains a total of 19 questions, which combine to form seven areas with corresponding scores, each showing a range between 0 and 3 points. In all cases, a score of “0” indicates ease, while a score of 3 indicates severe difficulty, within its respective area. The scores of the seven areas are finally summed to give an overall score, which ranges from 0 to 21 points. “0” indicates ease of sleeping and “21” indicates severe difficulty in all areas. The seven areas analyzed are Subjective Sleep Quality, Sleep Latency, Sleep Duration, Habitual Sleep Efficiency, Sleep Disturbances, Use of Sleep Medication, and Daytime Dysfunction [13].

The GHQ-12 is a mental health screening test, containing 12 questions about one’s health in the last few weeks. The scores on each of the items are added together. The higher the score, the greater the degree of emotional symptomatology. Scores of 12 or higher indicate the possibility that the person is suffering from an emotional disorder [14]. This cutoff has also been applied in previous studies involving university student populations, where it has shown good discriminative capacity for detecting non-psychotic psychiatric disorders or high levels of psychological distress [15].

HGS measurements were performed using a hydraulic hand dynamometer (JAMAR^®^ Hydraulic Hand Dynamometer, Model 5030J1; Performance Health Supply, Inc., Warrenville, IL, USA). HGS was assessed for both hands. Three measurements were taken 1 min apart and the highest determination was recorded. All measurements were conducted by trained physical therapy staff under standardized conditions. Participants were asked to indicate their dominant hand. In cases where participants reported equal use of both hands or ambidexterity, the hand used to write was considered dominant for classification purposes. Participants were seated comfortably with the elbow flexed at approximately 90 degrees, the forearm in a neutral position, and the wrist in a neutral position (between 0° and 30° extension and 0°and 15° ulnar deviation). This posture was maintained during all measurements to ensure consistency and reproducibility. While a formal standardized protocol was not applied, the procedure followed was consistent with the methodology described in the previous literature [5]. Additionally, the grip size of the JAMAR^®^ dynamometer was manually adjusted to fit each participant’s hand size to ensure a proper and comfortable grip. Although measurements were not limited to a specific time of day, all evaluations were conducted during regular university hours (between 8:30 a.m. and 3:30 p.m.) in a controlled and quiet environment. Participants were not instructed to fast, and no specific control over rest or fatigue levels was implemented. However, all measurements were performed under similar ambient conditions and supervised by trained staff to reduce variability as much as possible. Evaluations were performed individually in a quiet room, with each session lasting approximately 10–15 min per participant. Although data collection occurred over several days, the measurement conditions (environment, posture, time of day) were kept consistent to minimize variability.

### 2.2. Statistical Analysis

The responses obtained through the Google Forms^®^ platform will be collected in Excel format and subsequently transferred to the IBM^®^ SPSS^®^ Statistics software (IBM Corporation, headquartered in Armonk, NY, USA) in version 28.0 for statistical analysis. Quantitative variables will be defined by mean and standard deviation. Qualitative variables will be defined by number and percentage of cases. The sample size was determined based on the feasibility of recruitment during the academic year 2024–2025 and by reviewing similar cross-sectional studies conducted in university student populations. Previous research using comparable methodologies and variables reported sample sizes ranging from 80 to 150 participants [6]. Taking these studies as a reference, and considering the available population from the Physical Therapy program, a convenience sample was established. This approach ensured an adequate number of participants while maintaining methodological consistency with the existing literature.

The chi-square test will be used to study the relationship between qualitative variables. For the comparison of means, the Student’s *t*-test will be used for independent data (or the Mann–Whitney test when the quantitative variable does not follow a Gaussian distribution). The chi-square test was applied to evaluate associations between categorical variables due to its ability to compare proportions in independent groups. Spearman’s correlation test was used for the comparison of two quantitative variables. Spearman’s correlation was chosen due to the possible non-normal distribution of the data. Values for *p* < 0.05 will be considered statistically significant.

Cohen’s d values were calculated to assess the magnitude of the difference between two groups or conditions. It was used for interpreting whether an observed difference is significant from a practical point of view, beyond statistical significance. Values around 0.2 were considered as small effect size, values around 0.5 as moderate effect size, and values of 0.8 or more as large effect size.

### 2.3. Ethical Aspects

The Research Ethics Committee of Comillas Pontifical University reviewed and approved the study (protocol nº 27/24-25, date of approval 25 November 2024). The database was anonymized following the criteria set by Organic Law 3/2018 on Data Protection and in accordance with Regulation (EU) 2016/679, ensuring that participants cannot be directly or indirectly identified and that their data was processed securely. Furthermore, the study adhered to the ethical principles outlined in the Declaration of Helsinki, specifically those adopted at the 64th General Assembly in Fortaleza (Brazil), as well as current national legislation governing the analysis, protection, and confidentiality of personal data.

## 3. Results

A total of 145 students were enrolled, comprising 85 males (58.6%) and 60 females (41.4%) with a mean age of 21.0 ± 3.9 years. Sociodemographic and academic data are summarized in Table 1.

53 participants (36.6%) were in their first year of studies, 34 (23.4%) in their second year, 35 (24.1%) in their third year, and 23 (15.9%) in the fourth year. As personal history, 5 participants were diagnosed with asthma and 2 with migraines.

### 3.1. Assessment of Physical Activity

According to the results obtained by the IPAQ-SF questionnaire, the mean METS were 4634.0 ± 3204.2. By classifying them into categories, 123 participants (84.8%) performed vigorous activities (Table 1).

### 3.2. Pittsburgh Sleep Quality Index (PSQI)

The sum of the values of all the items gave a mean value of 6.9 ± 3.2. The separate results from the different domains of the PSQI are shown in Table 2. It is remarkable that the subjective sleep quality was perceived as quite bad or very bad in 27.5% of the subjects. Considering insufficient sleep as less than 7 h of sleep, we observed that 84.1% of the participants were within this range. In addition, the efficiency of sleep was below 85% in 26.2% of the subjects. Furthermore, 26.9% of the students take medication to sleep frequently and 11% weekly. Finally, 37.9% of the students reported moderate or severe dysfunctions in their usual activities during the day (Table 2).

### 3.3. Global Health Questionnaire (GHQ-12)

The sum of the values of all the questions gave a mean value of 11.4 ± 5.7. Considering scores of 12 or higher as indicators of the possibility that the person is suffering from an emotional disorder, 46 participants (31.7%) showed scores of over 12 points. The distribution of values is shown in Figure 1.

### 3.4. Hand Grip Strength Assessed by Dynamometer

Regarding laterality, 137 students were right-handed, 6 were left-handed, and 2 were ambidextrous. The mean HGS in the dominant hand was 42.4 ± 14.7 Kg. The average HGS in the non-dominant hand was 39.2 ± 14.8 Kg.

### 3.5. Association Between Sleep Quality and Hand Grip Strength

A significant inverse correlation could be established between total PSQI score and HGS (Spearman −0.178; *p* = 0.049) in the dominant hand (Spearman −0.211; *p* = 0.019) and the non-dominant hand.

### 3.6. Association Between Global Health Status and Hand Grip Strength

Those subjects with GHQ-12 scores at risk of emotional disorders (Score ≥ 12), showed significantly lower HGS values, both in the dominant hand (Mean difference 6.3 Kg; 95%CI (0.8–11.1); *p* = 0.023) and the non-dominant hand (Mean difference Kg; 95%CI (0.2–10.6); *p* = 0.042) (Table 3). In both the dominant and non-dominant hands, the effect size was mild to moderate.

### 3.7. Association Between Intensity of Physical Activity and Hand Grip Strength

It could be assumed that HGS could be highly influenced by the intensity of physical activity. However, we failed to demonstrate this association in both the dominant (*p* = 0.216) and non-dominant hand (*p* = 0.382).

### 3.8. Association Between Sleep Quality and Global Health Status

A significant direct correlation could be established between total PISQ and GHQ-12 scores (Spearman 0.514; *p* < 0.001). In those subjects with GHQ-12 scores under 12, mean PISQ values were 5.3 ± 2.5, while subjects with GHQ-12 scores over 12 showed mean PISQ values of 8.3 ± 3.4 (Mean difference 3.0; 95%CI (1.8–4.3; *p* < 0.001); Cohen’s d value was 0.94, representing a large effect size.

Table 4 shows a correlation matrix showing the relationships between hand grip strength, sleep indices, perception of health, and physical activity.

## 4. Discussion

The results of the present study suggest that students pursuing a Physical Therapy degree have a fair-to-poor sleep quality, and as a consequence, their functions are affected the following day in a significant number of cases. In addition, more than 30% of the students had scores of above 12 on the GHQ-12, suggesting the possibility of an emotional disorder. Lower values of hand grip strength were associated with sleep disturbances and elevated GHQ-12 scores, regardless of the intensity of physical activity.

The studied sample of students pursuing a Physical Therapy degree show an outstanding interest in physical activity, which translates into the fact that most of them (84.8%) can be included within the vigorous activity group. This makes the sample of these students a homogeneous and representative sample of a population of students with high physical activity. These data are in agreement with those described in another study carried out in another Spanish university, where the mean physical activity among Physical Therapy students was significantly higher than in other studies in the field of health sciences, such as medicine or nursing [15].

Within the young adult population, higher levels of physical activity are usually associated with healthier lifestyle habits, such as a healthier diet and lower consumption of toxic substances like alcohol, tobacco, or drugs. This would suggest that this population group should have a healthier perception of life, both physically and psychologically [16,17,18]. However, it is remarkable that 31.7% of our studied population showed scores of over 12 points, indicating that they are subjects at risk of suffering from an emotional disorder. The rising rates of depression have highlighted the growing worries about the mental health of university students in today’s academic setting. A recently published study among university students in Bangladesh, using the GHQ-12 questionnaire, revealed that around 60% of them suffer from mental disorders, mostly depression. The academic and social pressures that surround university students can significantly affect their mental health and outweigh the benefits of a healthy lifestyle [19]. A study conducted on Spanish university students pursuing health science degrees showed that one-third of the students reported having poor or very poor health, data close to those of our series, and in a population with lower levels of physical activity. However, it must be taken into consideration that in this study, health status was not assessed using the GHQ-12 questionnaire, but just via the answer to the question: “Within the last 12 months, how do you rate your overall health status?” [6]. Thus, the correlation of results must be interpreted with caution. The GHQ-12 is widely recognized as a valid and reliable tool for assessing psychological health. It is designed to screen for general psychiatric morbidity, such as anxiety, depression, and social dysfunction, in both clinical and non-clinical populations. The GHQ-12 has been extensively validated across diverse populations and cultural contexts. Studies have shown it has high internal consistency (Cronbach’s alpha above 0.8), indicating reliability. Furthermore, it has demonstrated good sensitivity and specificity in detecting psychological distress, making it effective for screening purposes. Its brevity (12 items) makes it practical for large-scale surveys and clinical settings, reducing respondent burden while maintaining accuracy, and it has been translated into multiple languages and adapted for various cultural settings, further supporting its validity as a universal screening tool [20].

Similarly, when analyzing the quality of sleep, it is noteworthy in our population of physical therapy students that 27.5% of them perceive their sleep quality as poor or very poor, that 84.1% sleep less than 7 h, that 26.9% of them have to resort frequently to medication to sleep, and that 37.9% describe that they are tired and cannot adequately perform their daily tasks. From these data, the most striking fact is the high percentage of students who sleep less than 7 h, which has been considered by other authors as insufficient sleep and a determinant cause of the development of physical and mental disorders [21,22]. Moreover, to achieve these hours of sleep, many students resort to the use of hypnotic drugs, with the consequent risk of addiction, especially if their consumption begins at such a young age. The most commonly used sleeping medications are benzodiazepines, and interestingly, the percentage of students who use them is higher than reported in other countries (4–7.9%) [23]. In the present study, a low quality of sleep determined by less than 7 h of sleep, along with other features of sleep disturbances, was associated with lower HGS. Several authors have described inadequate sleep as less than 7 h or more than 9 h [6]. Among young adults, a reduction in sleep duration is probably the most frequent disturbance. Due to the cross-sectional nature of the study, we cannot establish causal relationships between HGS and sleep quality.

HGS is a widely recognized marker of sarcopenia, which is the age-related loss of muscle mass and function. It is considered a reliable and simple measure to assess muscle strength and overall physical performance in older adults. Studies have shown that reduced HGS is associated with an increased risk of adverse health outcomes, including disability, falls, and mortality. As such, it serves as an important indicator for early detection and intervention of sarcopenia, helping healthcare professionals to implement appropriate strategies to maintain muscle health and improve quality of life [18]. Moreover, HGS has been identified as a predictor of various physical pathologies beyond sarcopenia. Diverse studies have demonstrated its association with cardiovascular diseases, metabolic disorders, and other chronic conditions. The simplicity and non-invasiveness of HGS measurement make it a valuable tool in clinical settings for assessing overall health and predicting the risk of developing serious health issues. By regularly monitoring HGS, healthcare providers can better identify individuals at risk and take preventive measures to mitigate potential health problems [24,25].

Despite most studies on HGS having been focused on physical pathologies, HGS has also emerged as a valuable marker for assessing psychological pathologies. Lower HGS has been associated with higher levels of depression, anxiety, and stress. This relationship is thought to be due to the physiological and psychological stress that can weaken muscle strength. Additionally, HGS has been linked to cognitive decline and dementia, suggesting that it could serve as an early indicator of these conditions [26]. Moreover, HGS can be a useful tool in clinical settings to monitor the mental health of patients. It provides a non-invasive and cost-effective method for assessing psychological well-being. Studies have indicated that HGS can predict the onset of mental health disorders and track the progress of treatment interventions. Therefore, incorporating HGS measurements into routine health assessments could enhance the early detection and management of psychological pathologies [27].

Some studies have also shown that HGS is associated with psychological health in university students. Jiang et al. found that greater HGS was linked to better cognitive functioning, higher life satisfaction, and reduced symptoms of depression and anxiety. This suggests that HGS could serve as an early indicator of mental health issues, allowing for timely interventions [26]. Tanaka et al. demonstrated that HGS predicts quality of life in both physical and psychological domains. This highlights the importance of HGS as a non-invasive measure for assessing mental well-being in young adults [28]. In our study, we observed significantly lower HGS values, both in the dominant and non-dominant hand, in subjects with GHQ-12 scores of over 12, indicating a potential risk of mental disorders. This is consistent with what has been previously described in the literature and emphasizes the importance of using dynamometry as a tool for early diagnosis and initial phases of psychological disorders that might otherwise go unnoticed.

Referring to the association between sleep disorders and HGS, evidence in the literature has shown that older adults with long sleep durations (≥9 h) had weaker HGS compared to those with mid-range sleep durations (7–8 h). This suggests that abnormal sleep patterns can negatively impact muscle strength and overall physical health [29]. Liu et al. reported similar findings, indicating that long sleep duration was associated with a higher prevalence of low grip strength [30]. In contrast, young adults and adolescents with shorter sleep durations had weaker HGS compared to those with adequate sleep [31]. In the present study, a low quality of sleep determined by less than 7 h of sleep, along with other features of sleep disturbances, was associated with lower hand grip strength. Several authors have described inadequate sleep as less than 7 h or more than 9 h. Among young adults, a reduction in sleep duration is probably the most frequent disturbance [6,22].

Lifestyle factors such as alcohol consumption or caffeine intake, which have not been recorded in the present study, might potentially affect both sleep and mental health. Alcohol consumption can significantly impact mental health and sleep patterns. It may initially promote relaxation, but excessive intake can lead to increased anxiety and depression, as alcohol disrupts neurotransmitter balance in the brain. It also interferes with sleep quality by suppressing REM (rapid eyes movements) sleep, resulting in fatigue and reduced cognitive function the next day. Over time, habitual alcohol use can contribute to insomnia, creating a cycle of poor sleep and declining mental well-being [32]. On the other hand, caffeine consumption has notable effects on mental health and sleep, acting as a stimulant that temporarily boosts alertness, mood, and energy by blocking adenosine, a neurotransmitter responsible for promoting sleepiness. While moderate intake is generally considered safe and even beneficial for focus and productivity, excessive consumption can lead to heightened anxiety, restlessness, and irritability, particularly in individuals sensitive to caffeine. Regarding sleep, caffeine can disrupt sleep patterns by delaying the onset of sleep, reducing overall sleep duration, and decreasing the quality of restorative deep sleep, especially if consumed later in the day [33].

Finally, it could be expected that HGS would be associated with the intensity of physical activity [31]. However, there are other factors to consider, such as the type of physical activity. Sports that require greater grip strength in the hands will show higher HGS values. However, this may not be the case for individuals who practice sports that exercise the back or lower body. The type of physical activity our participants engaged in was so heterogeneous that we could not find significant differences in favor of any specific type of sports practice.

### Limitations and Strengths

The first limitation of this study is that the sample is limited to students pursing a Physical Therapy degree and to students with high levels of physical activity, possibly associated with healthier lifestyle habits. Therefore, these results cannot be extrapolated to the entire young adult population, or even to the entire university community. The use of non-probabilistic sampling introduces a selection bias that limits the generalizability of the results. In addition, the sample size is relatively small, as the study population was only limited to one university degree, aiming to homogenize the sample. This also limits the external validation of the results. Moreover, multivariate analysis showed that the PISQ score was the only variable that was independently associated with the GHQ-12 score. For the other associations, statistical significance was not achieved, probably due to the small sample size. Further studies must be conducted with a higher number of participants and including other university degrees.

Furthermore, the quality of sleep was determined by means of a questionnaire with a certain degree of subjectivity. There are other more objective measures, like different types of sleep monitoring (i.e., polysomnography), that should be probably employed in future studies. The cross-sectional design does not allow us to determine whether the reduction in HGS precedes sleep disturbances or vice versa.

In addition, mental health was assessed with the GHQ-12 questionnaire. This can be a useful screening tool, but there are other more precise questionnaires able to determine if the mental disorders are more focused on depression, anxiety, or stress, as well as identify which are the more frequent mental disorders among university students. Future studies should also implement these more specific questionnaires for a better characterization of the disorders.

The small sample size has also prevented conducting a regression model to assess whether the association between HGS and perceived mental health varies depending on sleep quality, or vice versa. Future research must consider the possibility of analyzing this interaction.

Finally, as previously mentioned, lifestyle factors such as alcohol consumption, smoking, diet, or caffeine intake, which can be potential confounders, have not been recorded in the present study. These factors must be taken into consideration for future studies in order to determine the impact that they can have on health status, sleep disturbances, and HGS.

On the other hand, the main strengths of the present study are that it has been conducted on a homogeneous sample of university students with vigorous activity habits. Furthermore, this study has shown the relationship of HGS with psychological disorders and sleep disturbances in young adults with high educational levels, which has previously little academic focus. Although the sample size is not excessively large and the effect of the association is low-to-moderate, these results should highlight the important prevalence of psychological pathology and sleep disturbances in the university population and that a low HGS may be an early indicator of it. We consider that the results of the present study can be a starting point for future research projects in this line to confirm our results.

## 5. Conclusions

Students pursuing a Physical Therapy degree have a fair-to-bad quality of sleep, with more than 80% of them sleeping less than 7 h and often requiring medication for sleep. As a result, their functions are affected the next day in a significant number of cases. In addition, over 30% of the students presented scores over of 12 in the GHQ-12, suggesting the possibility of suffering from an emotional disorder. Lower hand grip strength values are associated with sleep disorders and elevated scores in the GHQ-12, independent of the intensity of physical activity. Our findings suggest that HGS could be an accessible marker of psychological risk in university students, although longitudinal studies are needed to confirm this hypothesis.

## Figures and Tables

**Figure 1 jfmk-10-00122-f001:**
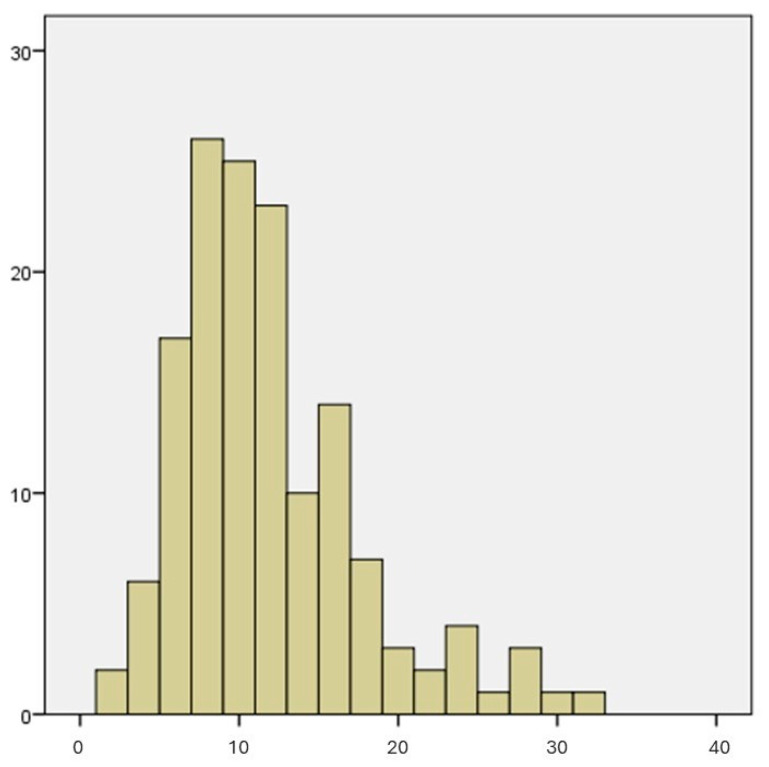
Histogram showing the GHQ-12 score distribution.

**Table 1 jfmk-10-00122-t001:** Sociodemographic and academic data of the sample and categories of physical activity.

**Age (years)**	21.0 ± 3.9
Gender -Males	85 (58.6%)
Study course -First -Second -Third -Fourth	53 (36.6%) 34 (23.4%) 35 (24.1%) 23 (15.9%)
Comorbidities -Asthma -Migraines	5 (3.4%) 2 (1.4%)
Physical activity	
-Sedentarism	3 (2.1%)
-Mild activity	6 (4.1%)
-Moderate activity	13 (9.0%)
-Vigorous activity	123 (84.8%)

**Table 2 jfmk-10-00122-t002:** Domains of the PSQI.

**Item 1: Subjective sleep quality**	**N**	**%**
Very good	21	14.5
Fairly good	84	57.9
Fairly bad	34	23.4
Very bad	6	4.1
**Item 2: Sleep latency**	**N**	**%**
0.00	33	22.8
1.00	68	46.9
2.00	27	18.6
3.00	17	11.7
**Item 3: Sleep duration**	**N**	**%**
>7 h	23	15.9
6–7 h	71	49.0
5–6 h	35	24.1
<5 h	16	11.0
**Item 4: Sleep efficiency**	**N**	**%**
>85%	93	73.8
75–84%	23	18.3
65–74%	6	4.8
<65%	4	3.2
**Item 5: Sleep perturbances**	**N**	**%**
0.00	5	3.4
1.00	121	83.4
2.00	19	13.1
**Item 6: Sleep medication**	**N**	**%**
Not at all in the last month	106	73.1
Less than 1 time/week	23	15.9
1–2 times/week	9	6.2
>3 times/week	7	4.8
**Item 7: Dysfunction during the day**	**N**	**%**
0.00	28	19.3
1.00	62	42.8
2.00	44	30.3
3.00	11	7.6

**Table 3 jfmk-10-00122-t003:** Association between GHQ-12 score and hand grip strength in the dominant and non-dominant hand.

	GHQ-12	N	Mean	Standard Deviation	*p*	Cohen’s d
**Dominant hand**	<12	96	44.3	15.3	0.023	0.4
	>12	46	38.3	12.6		
**Non-Dominant hand**	<12	96	40.9	15.5	0.042	0.36
	>12	46	35.5	12.7		

**Table 4 jfmk-10-00122-t004:** Correlation matrix showing the relationships between hand grip strength in dominant and non-dominant hands, sleep indices (PISQ), perception of health (GHQ-12), and physical activity (IPAQ).

		PISQ	GHQ-12	IPAQ	HGS Dominant Hand	HGS Non-Dominant Hand
PISQ	Spearman	1	0.514	0.012	−0.178	−0.211
	*p* value		<0.001	0.891	0.049	0.019
GHQ-12	Spearman	0.514	1	−0.071	−0.249	−0.246
	*p* value	<0.001		0.399	0.003	0.003
IPAQ	Spearman	0.012	−0.071	1	0.145	0.134
	*p* value	0.891	0.399		0.216	0.382
HGS dominant hand	Spearman	−0.178	−0.249	0.145	1	0.935
	*p* value	0.049	0.003	0.216		<0.001
HGS non-dominant hand	Spearman	−0.211	−0.246	0.134	0.935	1
	*p* value	0.019	0.003	0.382	<0.001	

Multivariate analysis showed that the PISQ score was the only variable that was independently associated with the GHQ-12 score (*p* < 0.001). For the other associations, statistical significance was not achieved.

## Data Availability

Data will be available on request to the authors.

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
