# Peer review of "Correlation of Hand Grip Strength with Sleep Quality and Perception of General Health Status in University Students: A Cross-Sectional Study"

_jfmk, 2025, doi:10.3390/jfmk10020122_

Round 1
Reviewer 1 Report
Comments and Suggestions for Authors
First of all, I would like to congratulate the authors on their work and their contribution to this research. Writing any scientific document requires considerable effort.
However, the study presents a series of aspects that should be addressed before making a final decision:
TITLE
• I believe it is important and necessary to specify the study population in the title.
ABSTRACT
• The abstract contains 330 words, and I consider this to be somewhat excessive in terms of the information provided. (I suggest that the authors check whether there is a word limit for the abstract in the journal's guidelines). If there is no limit, I still believe it could be made more concise.
• Given how the title and objective have been formulated (first mentioning strength and then the other variables), this should be taken into account as a starting point in the scientific writing of the document. In the case of the abstract, this should be applied when mentioning the variables used, as well as when presenting the results of the different variables, beginning with the strength data first and then the rest. The same applies to the conclusion—if the study focuses on handgrip strength in relation to sleep quality and health status, it seems inconsistent for the conclusion to begin by mentioning sleep data instead of handgrip strength.
• Line 17: Insert a space before the word "results."
INTRODUCTION
• I would indicate the reference for this paragraph, regardless of whether the following paragraph cites the same reference:
Gomez-Campos et al. have shown that there is a nonlinear relationship between chronological age and HGS from childhood to senescence, with men experiencing an accelerated and continuous reduction in HGS and women experiencing a reduction in HGS from the age of 40 years.
• This paragraph contains some aspects that should be considered:
Sleep disturbances have also been associated with reduced HGS and sarcopenia in the elderly, due to low muscle strength or poor physical performance, showing a direct association with poor quality of life [PLEASE INSERT REFERENCE]. Sleep plays an important role in the functioning of cells, organs, and systems. Several studies have shown that altered sleep duration is associated with an increased risk of mortality in middle and old age. Insufficient sleep can reduce the level of growth hormone (GH) and increase the level of cortisol and proinflammatory changes, which in turn increase the risk of cardiovascular disease, insulin resistance, diabetes, and obesity [PLEASE INSERT REFERENCE]. Several studies have suggested that both short sleep duration and an excessive number of hours of sleep (both situations encompassed within the concept of pathological sleep) have a positive association with sarcopenia [7-12].
What specific disease? Sarcopenia? It is not clear to the reader:
Although there is little evidence in the literature regarding the association between sleep disturbances and HGS in young adults, it is possible to find incipient associations in this age range that should be considered as potential markers of future physical and psychological pathologies, allowing early intervention to modify the progression of the disease [1-3].
• I believe the paragraph order should be adjusted based on the sequence in which the elements are introduced. It would be preferable to first state that lower arm strength levels may be associated with lower levels of… (see attached paragraph).
Based on previous literature, we hypothesize that lower sleep quality and poorer perceived health will be associated with lower HGS in young adults. The aims of this study were to establish the relationship between HGS and sleep disturbances and to correlate HGS with the perception of general health status.
MATERIALS AND METHODS
• Line 83: There is a double space before the word "exclusion."
• Recent anatomical lesions—but within what time frame?
• Who conducted the assessments? Were all assessments performed at the same time, even if on different days? How long did the evaluations take?
• Consider adding the number of participants again in this paragraph:
A cross-sectional observational study was conducted. Consecutive non-probability sampling was performed for the recruitment of participants from the Physical Therapy degree at the San Rafael campus of the San Juan de Dios School of Nursing and Physical Therapy, Comillas Pontifical University, during the academic year 2024-2025. Exclusion criteria included a personal history of neuromuscular pathology affecting the determination of HGS, current or recent anatomical lesions hindering hand mobility, and sequelae of previous trauma, surgery, or congenital alterations affecting HGS assessment.
• If a student was an Erasmus student, for example, an English-speaking student, were they excluded despite being a physical therapy student? Or were the questionnaires adapted? This should be considered in the inclusion/exclusion criteria.
• If a participant considers both hands dominant or can write equally well with both, which hand is classified as dominant?
• Regarding the questionnaires, I would provide the reliability or validity (Intraclass Correlation Coefficient or Cronbach’s Alpha) of the tool, along with bibliographic references, ideally in the studied population.
• If participants had different hand sizes, is there a protocol specifying the hand position for the grip measurement before the participant squeezes? It would be important to mention this.
• If the study is based on previous research, references should be provided. Additionally, I suggest writing a more specific paragraph explaining and justifying the sample size:
The sample size was based on similar previous studies and the feasibility of recruitment during the academic period.
• If possible, I would include Cohen’s d for the comparisons and mention it in the analysis, results, and relevant tables.
RESULTS
• Is there a table that provides all this information, including demographic characteristics, asthma, etc., of the sample?
A total of 145 students were enrolled: 85 males (58.6%) and 60 females (41.4%), with a mean age of 21.0 ± 3.9 years. Among them, 53 participants (36.6%) were in their first year of studies, 34 (23.4%) in their second year, 35 (24.1%) in their third year, and 23 (15.9%) in their fourth year. Regarding personal medical history, 5 participants were diagnosed with asthma, and 2 with migraines.
• The p in the p-value should be italicized; ensure consistency throughout the document and tables.
• The standard deviation in Table 3 is presented with four decimal places, while the other tables and figures typically use three decimals in the document. I suggest unifying this aspect.
DISCUSSION
• The first paragraph of the discussion does not align with how the objectives and variables are presented in the document.
• Lines 194-197: It is mentioned that the findings align with those of other studies, but only one reference is provided. Either more references should be included, or the statement should be in the singular form.
• Lines 210-213: How was poor health determined in that study? Was the same tool used as in this study, or was a different instrument employed? It would be interesting to mention this.
• Lines 227-228 should include a reference.
• Lines 238, 245, and 249: Separate HGS from the word has, and in lines 250 and 253, separate HGS from can.
• Separate HGS from as in line 262.
• Separate HGS from values in line 264.
• I suggest reviewing the entire discussion section to correct these recurring writing errors.
• The Limitations section should be accompanied by a Strengths section, where the positive aspects, relevance, and clinical implications of this study are highlighted.
CONCLUSION
• Reformulate the conclusion based on all the aforementioned changes, improving the writing by avoiding excessive paragraph breaks and unifying ideas.
Comments on the Quality of English Language
he writing and English of the document should be reviewed
Author Response
REVIEWER 1:
First of all, I would like to congratulate the authors on their work and their contribution to this research. Writing any scientific document requires considerable effort.
However, the study presents a series of aspects that should be addressed before making a final decision:
TITLE
• I believe it is important and necessary to specify the study population in the title.
ANSWER: We appreciate the reviewer’s insightful suggestion. In response, we have revised the title to clearly specify the study population. The updated title now reads:
“Correlation of hand grip strength with sleep quality and perceived general health status in university students: a cross-sectional study.”
ABSTRACT
- The abstract contains 330 words, and I consider this to be somewhat excessive in terms of the information provided. (I suggest that the authors check whether there is a word limit for the abstract in the journal's guidelines). If there is no limit, I still believe it could be made more concise.
• Given how the title and objective have been formulated (first mentioning strength and then the other variables), this should be taken into account as a starting point in the scientific writing of the document. In the case of the abstract, this should be applied when mentioning the variables used, as well as when presenting the results of the different variables, beginning with the strength data first and then the rest. The same applies to the conclusion—if the study focuses on handgrip strength in relation to sleep quality and health status, it seems inconsistent for the conclusion to begin by mentioning sleep data instead of handgrip strength.
• Line 17: Insert a space before the word "results."
ANSWER: We thank the reviewer for these constructive comments, which have helped improve the clarity, conciseness, and structure of the abstract.
- In response to the suggestion regarding word count, we have revised the abstract to reduce it from 330 words to approximately 230. While the journal does not specify a strict word limit, we agree that a more concise version improves readability and focus.
- Following the recommendation to align the abstract with the structure of the title and objectives, we have reorganized the content to prioritize hand grip strength (HGS) as the main variable of interest. HGS now appears first when describing the aims, variables, results, and conclusions, reinforcing the central focus of the study.
- We have also corrected the formatting issue indicated in line 17 by inserting the missing space before the word “results.”
INTRODUCTION
- I would indicate the reference for this paragraph, regardless of whether the following paragraph cites the same reference:
Gomez-Campos et al. have shown that there is a nonlinear relationship between chronological age and HGS from childhood to senescence, with men experiencing an accelerated and continuous reduction in HGS and women experiencing a reduction in HGS from the age of 40 years.
ANSWER: We appreciate the reviewer’s attention to detail. In response to this suggestion, we have added the corresponding citation [5] immediately following the first mention of Gomez-Campos et al. in the paragraph, even though the reference already appeared at the end of the paragraph. This change improves the clarity and traceability of the source.
- This paragraph contains some aspects that should be considered:
Sleep disturbances have also been associated with reduced HGS and sarcopenia in the elderly, due to low muscle strength or poor physical performance, showing a direct association with poor quality of life [PLEASE INSERT REFERENCE]. Sleep plays an important role in the functioning of cells, organs, and systems. Several studies have shown that altered sleep duration is associated with an increased risk of mortality in middle and old age. Insufficient sleep can reduce the level of growth hormone (GH) and increase the level of cortisol and proinflammatory changes, which in turn increase the risk of cardiovascular disease, insulin resistance, diabetes, and obesity [PLEASE INSERT REFERENCE]. Several studies have suggested that both short sleep duration and an excessive number of hours of sleep (both situations encompassed within the concept of pathological sleep) have a positive association with sarcopenia [7-12].
What specific disease? Sarcopenia? It is not clear to the reader:
Although there is little evidence in the literature regarding the association between sleep disturbances and HGS in young adults, it is possible to find incipient associations in this age range that should be considered as potential markers of future physical and psychological pathologies, allowing early intervention to modify the progression of the disease [1-3].
ANSWER: Thank you for these valuable observations. In response, we have made the following corrections:
- We added the appropriate references to support the associations between sleep disturbances, HGS, sarcopenia, and the hormonal and metabolic consequences of poor sleep.
- We replaced the term “elderly” with “older adults” and “older individuals” to ensure the use of respectful and non-ageist language.
- We clarified the reference to “the disease” at the end of the paragraph by explicitly mentioning sarcopenia and other possible future conditions such as psychological disorders, improving the precision of the statement.
- The paragraph has also been revised to enhance scientific clarity and alignment with the rest of the manuscript.
- I believe the paragraph order should be adjusted based on the sequence in which the elements are introduced. It would be preferable to first state that lower arm strength levels may be associated with lower levels of… (see attached paragraph).
Based on previous literature, we hypothesize that lower sleep quality and poorer perceived health will be associated with lower HGS in young adults. The aims of this study were to establish the relationship between HGS and sleep disturbances and to correlate HGS with the perception of general health status.
ANSWER: Thank you for this helpful suggestion. We have revised the final paragraph of the introduction to better reflect the logical progression of the manuscript, starting with hand grip strength as the main outcome variable. The updated text now reads:
"Based on previous literature, we hypothesize that lower hand grip strength (HGS) will be correlated with lower sleep quality and poorer perceived general health in young adults. The aims of this study were to examine the relationship between HGS and sleep disturbances, and to explore the association between HGS and perceived general health status."
MATERIALS AND METHODS
- Line 83: There is a double space before the word "exclusion."
ANSWER: Thank you for pointing this out. The extra space before the word “exclusion” has been corrected in the revised manuscript.
Recent anatomical lesions—but within what time frame?
ANSWER: We appreciate this observation. Based on commonly used criteria in the literature, we have now specified the time frame as “within the past three months” to clearly define what constitutes a recent anatomical lesion that could impact hand mobility and HGS assessment.
Who conducted the assessments? Were all assessments performed at the same time, even if on different days? How long did the evaluations take?
ANSWER: Thank you for your detailed and relevant comment. We have added information clarifying that all HGS assessments were performed by trained physical therapy staff in standardized conditions. Each evaluation session lasted approximately 10–15 minutes and was conducted individually in a controlled environment. Although the data collection was spread over multiple days, all conditions (location, measurement protocol, and time of day) were kept consistent to minimize variability.
Consider adding the number of participants again in this paragraph:
A cross-sectional observational study was conducted. Consecutive non-probability sampling was performed for the recruitment of participants from the Physical Therapy degree at the San Rafael campus of the San Juan de Dios School of Nursing and Physical Therapy, Comillas Pontifical University, during the academic year 2024-2025. Exclusion criteria included a personal history of neuromuscular pathology affecting the determination of HGS, current or recent anatomical lesions hindering hand mobility, and sequelae of previous trauma, surgery, or congenital alterations affecting HGS assessment.
ANSWER: Thank you for your suggestion. We have now added the number of participants in the paragraph describing the study design and recruitment process to improve clarity and provide immediate context regarding the study sample.
- If a student was an Erasmus student, for example, an English-speaking student, were they excluded despite being a physical therapy student? Or were the questionnaires adapted? This should be considered in the inclusion/exclusion criteria.
ANSWER: We thank the reviewer for highlighting this important point. We confirm that the questionnaires used (IPAQ-SF, PSQI, GHQ-12) were only available and validated in Spanish. Therefore, students who were not proficient in Spanish were excluded from the study to ensure accurate comprehension and response validity. This detail has now been explicitly included in the exclusion criteria.
If a participant considers both hands dominant or can write equally well with both, which hand is classified as dominant?
ANSWER: Thank you for this observation. We have clarified in the methodology that participants were asked to report their dominant hand, and in cases of ambidexterity or equal use, the hand used for writing was considered dominant. This information has now been added to the HGS assessment section.
Regarding the questionnaires, I would provide the reliability or validity (Intraclass Correlation Coefficient or Cronbach’s Alpha) of the tool, along with bibliographic references, ideally in the studied population.
ANSWER: Thank you for this helpful suggestion. We have added information regarding the reliability and validity of the three instruments used in the study (IPAQ-SF, PSQI, GHQ-12). For the IPAQ-SF, we included reliability data specifically from studies conducted in university student populations. For the PSQI and GHQ-12, although direct validations in university samples are limited, we referenced values of internal consistency reported in general or Spanish-speaking adult populations. All corresponding references have been added to the revised manuscript.
If participants had different hand sizes, is there a protocol specifying the hand position for the grip measurement before the participant squeezes? It would be important to mention this.
ANSWER: We thank the reviewer for this pertinent comment. We have clarified in the methodology section that, although a standardized protocol was not formally applied, the procedure followed was consistent with that described in previous literature. Participants were seated with the elbow flexed at approximately 90 degrees and the wrist in a neutral position, and the grip size of the JAMAR® dynamometer was manually adjusted to accommodate individual hand sizes.
If the study is based on previous research, references should be provided. Additionally, I suggest writing a more specific paragraph explaining and justifying the sample size:
The sample size was based on similar previous studies and the feasibility of recruitment during the academic period.
ANSWER: Thank you for this important suggestion. We have revised the paragraph to include a more specific justification for the sample size, referencing previous studies with similar variables and target populations. The new paragraph also clarifies that the final sample was determined based on recruitment feasibility during the academic period and aligned with comparable published research.
- If possible, I would include Cohen’s d for the comparisons and mention it in the analysis, results, and relevant tables.
ANSWER: Cohen´s d values were calculated to assess the magnitude of the difference between two groups or conditions. It was used for interpreting whether an observed difference is significant from a practical point of view, beyond statistical significance. Values around 0.2 were considered as small effect size, values around 0.5 as moderate effect size and values of 0.8 or more as large effect size. (Statistical analysis section, 3rd paragraph).
Cohen´s d values were calculated for the comparison of HGS with GHQ scores over or under 12 (Table 4).
RESULTS
- Is there a table that provides all this information, including demographic characteristics, asthma, etc., of the sample?
A total of 145 students were enrolled: 85 males (58.6%) and 60 females (41.4%), with a mean age of 21.0 ± 3.9 years. Among them, 53 participants (36.6%) were in their first year of studies, 34 (23.4%) in their second year, 35 (24.1%) in their third year, and 23 (15.9%) in their fourth year. Regarding personal medical history, 5 participants were diagnosed with asthma, and 2 with migraines.
ANSWER: All this information has been summarized in Table 1
- The pin the p-valueshould be italicized; ensure consistency throughout the document and tables.
ANSWER: p-values have been italicized thoughout the document and tables.
The standard deviation in Table 3 is presented with four decimal places, while the other tables and figures typically use three decimals in the document. I suggest unifying this aspect.
ANSWER: Decimal expression has been unified only to 1 decimal throughout the text, excepting p values that are expressed with three decimals
.
DISCUSSION
- The first paragraph of the discussion does not align with how the objectives and variables are presented in the document.
ANSWER: We agree with the reviewer observation. We have therefore added a first introductory paragraph to the Discussion highlighting the main results from the study. From there, we would like to highlight in the second paragraph that the level of physical activity in the sample studied is globally very high, which makes it a homogeneous sample in which the level of physical activity does not bias the results, although it does limit the external validity of the results. “ The results of the present study suggest that students of the Physical Therapy degree have a fair to poor sleep quality, and as a consequence, their functions are affected the following day in a significant number of cases. In addition, more than 30% of the students had scores above 12 on the GHQ-12, suggesting the possibility of an emotional disorder. Lower values of hand grip strength were associated with sleep disturbances and elevated GHQ-12 scores, regardless of the intensity of physical activity.” (Discussion, 1st paragraph)
Lines 194-197: It is mentioned that the findings align with those of other studies, but only one reference is provided. Either more references should be included, or the statement should be in the singular form.
ANSWER: We agree with the observation. The statement refers to only one study. Consequently, the sentence has been rewritten in singular form “These data are in agreement with those described in other study carried out in another Spanish university, where the mean physical activity among Physical Therapy students was significantly higher than in other studies in the field of health sciences, such as medicine or nursing [15].” (Discussion, 2nd paragraph)
Lines 210-213: How was poor healthdetermined in that study? Was the same tool used as in this study, or was a different instrument employed? It would be interesting to mention this.
ANSWER: The following sentences has been added to clarify that different tools were used to obtain the information about health status “A study conducted on Spanish university students pursuing health science degrees showed that one-third of the students reported having poor or very poor health, data close to those of our series, and in a population with lower levels of physical activity. However, it must be taken into consideration that in this study that health status was not assessed using the GHQ-12 questionnaire, but just via the answer to the question: “Within the last 12 months, how do you rate your overall health status?” [6]. Thus, the correlation of results must be interpreted with caution. ”
Lines 227-228 should include a reference.
ANSWER: A reference has been added to this statement “Several authors have described inadequate sleep as less than 7 hours or more than 9 hours [6]”
Lines 238, 245, and 249: Separate HGSfrom the word has, and in lines 250 and 253, separate HGSfrom can. • Separate HGS from as in line 262. Separate HGS from values in line 264.
ANSWER: All these spelling mistakes have been corrected.
I suggest reviewing the entire discussion section to correct these recurring writing errors.
ANSWER: the whole text has been revised by a native English speaker.
The Limitationssection should be accompanied by a Strengthssection, where the positive aspects, relevance, and clinical implications of this study are highlighted.
ANSWER: The Limitations section has been changed to “4.1 Limitations and strengths section”. The following paragraph has been added at the end of this section “On the other hand, the main strengths of the present study are that it has been conducted on a homogeneous sample of university students with vigorous activity habits. Furthermore, this study has shown the relationship of HGS with psychological disorders and sleep disturbances in young adults with high educational levels, which has been little studied previously. Although the sample size is not excessively large and the effect of the association low-to-moderate, these results should highlight the important prevalence of psychological pathology and sleep disturbances in the university population and that a low GSH may be an early indicator of it. We consider that the results of the present study can be a starting point for future research projects in this line to confirm our results.”
CONCLUSION
- Reformulate the conclusion based on all the aforementioned changes, improving the writing by avoiding excessive paragraph breaks and unifying ideas.
ANSWER: The Conclusion section has been reformulated, avoiding paragraph breaks and unifying ideas.
Reviewer 2 Report
Comments and Suggestions for Authors
Manuscript ID: jfmk-3519546
Manuscript title: Association of hand grip strength with sleep quality and perception of general health status: a cross-sectional study
Title section:
Comment 1: The study focuses on university students, but the title does not explicitly indicate this population. To enhance clarity and specificity, it might be helpful to include "in university students" in the title. This addition could make it easier for readers to immediately grasp the study’s target population.
Abstract section:
Comment 2: "Lower hand grip strength values are associated with sleep disorders and elevated scores in the GHQ-12." However, the phrase "associated with" might be misinterpreted as causation rather than correlation. To avoid confusion, it is recommended to use "correlated with" instead.
Introduction section:
Comment 3: The authors use the term "elderly" to refer to older individuals. This term is increasingly considered ageist and discriminatory. It is advisable to use alternative, more neutral terms such as "older adults" or "older people" to avoid perpetuating negative stereotypes and to promote respectful language throughout the manuscript.
Comment 4: While the introduction references numerous studies on older and middle-aged adults, this study specifically targets students. Although the authors acknowledge the scarcity of research on hand grip strength (HGS) in young adults, they fail to adequately justify the relevance of assessing HGS within a student population. It is essential to clarify why evaluating HGS in students is meaningful and what unique insights can be gained from this demographic to provide a solid rationale for the study.
Methods section:
Comment 5: The study design (cross-sectional study) is clearly described. However, it is not entirely clear how lifestyle factors such as alcohol consumption, smoking, diet, and caffeine intake were accounted for. Were any statistical adjustments, such as multivariate regression analysis, performed to control for these potential confounders? If not, discussing their possible impact on the study findings could enhance the robustness of the interpretation.
Comment 6: For HGS measurement, specifying the posture and hand position (e.g., seated or standing) would improve reproducibility.
Comment 7: The study does not appear to consider variations in HGS due to measurement conditions. HGS fluctuates depending on factors such as the time of day and testing environment. For example, grip strength tends to be lower in the morning and increases throughout the day. Additionally, factors such as whether the measurement was taken after eating or while fasting, and whether participants were well-rested or fatigued, could have influenced the results. Were the measurement conditions standardized to minimize these variations?
Comment 8: "The GHQ-12 is a mental health screening test." It would be beneficial to discuss whether this cutoff score is appropriate for the study population (university students) by comparing it with previous studies.
Comment 9: The analysis presented is insufficient and does not adequately explore the relationships between the variables. At a minimum, the following analyses should be included in this study:
Correlation analysis: A correlation matrix showing the relationships between hand grip strength, sleep indices, subjective perception of health, and other relevant factors.
Regression analysis: Univariate and multiple linear or logistic regression analyses should be performed, with subjective perception of health as the outcome variable. These analyses should compare the contribution of grip strength, sleep indices, and other relevant factors to the outcome. This will help determine the relative importance of each factor in predicting subjective health perception.
Comment 10: When explaining the relationship with subjective health perception, it is essential to consider the interaction effect between grip strength and sleep indices. Investigating whether the impact of grip strength on subjective health perception varies depending on the level of sleep quality, or vice versa, would provide a more nuanced understanding of their interplay. Such an interaction analysis is crucial for a comprehensive interpretation of the results.
Results section:
Comment 11: Table 1 currently presents only the results of physical activity. To provide a more comprehensive understanding of the participants' characteristics, it would be beneficial to include data on various other factors that define the characteristics of the subjects. By including a wider range of relevant variables, the table could offer a more detailed profile of the study population
Comment 12: Visualizing the GHQ-12 score distribution (e.g., using a histogram or box plot) would enhance clarity.
Comment 13: There are two sections labeled as '3.4' in the results section: '3.4. Hand grip strength assessed by dynamometer' and '3.4. Association between sleep quality and hand grip strength.' To ensure clarity and consistency, it would be advisable to renumber the second '3.4' appropriately.
Comment 14: Table 3 shows a statistically significant difference in HGS between males and females (p = 0.000). However, this result is biologically expected and does not provide novel insight. Since sex-based differences in muscle strength are well-established, emphasizing this finding as a key result may not add substantial value. Would it be more appropriate to shift the focus to other, more meaningful comparisons within the study?
Comment 15: In several instances, the paper reports p-values as "p = 0.000" (e.g., in Table 3 and related text in the Results section). This is incorrect because p-values represent probabilities and can never be exactly zero. Instead, the appropriate way to report these values is "p < 0.001", which follows scientific reporting standards.
Comment 16: The notation of p-values in the statistical results is inconsistent in terms of uppercase and lowercase letters (e.g., 'P' vs. 'p'). Ensuring uniform formatting throughout the manuscript would improve clarity and professionalism.
Discussion section:
Comment 17: The association between HGS and psychological health is discussed; however, further clarification on the validity of using GHQ-12 as a psychological health indicator would strengthen the argument.
Summary:
This study provides valuable insights into the relationship between hand grip strength, sleep quality, and perceived health status in university students. However, clarity, methodological rigor, and consistency need improvement. Key revisions should focus on title specificity, statistical control of lifestyle factors, standardization of HGS measurement conditions, and proper p-value reporting. Strengthening the logical flow and scientific precision will enhance the manuscript’s impact.
Author Response
REVIEWER 2:
Title section:
Comment 1: The study focuses on university students, but the title does not explicitly indicate this population. To enhance clarity and specificity, it might be helpful to include "in university students" in the title. This addition could make it easier for readers to immediately grasp the study’s target population.
ANSWER: Thank you for this helpful suggestion. We agree that including the target population improves the clarity and specificity of the title. Accordingly, we have revised the title to reflect this change. The updated title now reads:
"Correlation of hand grip strength with sleep quality and perceived general health status in university students: a cross-sectional study."
Abstract section:
Comment 2: "Lower hand grip strength values are associated with sleep disorders and elevated scores in the GHQ-12." However, the phrase "associated with" might be misinterpreted as causation rather than correlation. To avoid confusion, it is recommended to use "correlated with" instead.
ANSWER: Thank you for this valuable observation. We agree that the term “correlated with” is more appropriate given the cross-sectional design of the study. Accordingly, we have revised the sentence in the abstract to read:
“Lower hand grip strength values were correlated with poor sleep quality and higher GHQ-12 scores, independently of physical activity levels.”
Introduction section:
Comment 3: The authors use the term "elderly" to refer to older individuals. This term is increasingly considered ageist and discriminatory. It is advisable to use alternative, more neutral terms such as "older adults" or "older people" to avoid perpetuating negative stereotypes and to promote respectful language throughout the manuscript.
ANSWER: We thank the reviewer for this important and thoughtful comment. In response, we have revised the manuscript by replacing the term “elderly” with “older adults” in all relevant instances, to align with current standards for respectful and inclusive language in scientific writing.
Comment 4: While the introduction references numerous studies on older and middle-aged adults, this study specifically targets students. Although the authors acknowledge the scarcity of research on hand grip strength (HGS) in young adults, they fail to adequately justify the relevance of assessing HGS within a student population. It is essential to clarify why evaluating HGS in students is meaningful and what unique insights can be gained from this demographic to provide a solid rationale for the study.
ANSWER: We thank the reviewer for this important observation. In response, we have expanded the introduction to better justify the relevance of studying HGS in university students. We now highlight the transitional nature of this life stage, the influence of emerging lifestyle habits, and the potential value of HGS as an early health indicator in this demographic. This addition strengthens the rationale for focusing on a student population in the present study.
Methods section:
Comment 5: The study design (cross-sectional study) is clearly described. However, it is not entirely clear how lifestyle factors such as alcohol consumption, smoking, diet, and caffeine intake were accounted for. Were any statistical adjustments, such as multivariate regression analysis, performed to control for these potential confounders? If not, discussing their possible impact on the study findings could enhance the robustness of the interpretation.
ANSWER: These factors have not been recorded for the present study. Thus, we have included them as a Limitation of the study “Finally, lifestyle factors such as alcohol consumption, smoking, diet or caffeine intake, which can be potential confounders, have not been recorded in the present study. These factors must be taken into consideration for future studies in order to determine the impact that they can have on health status, sleep disturbances and HGS. ” (Limitations and strengths section, 4th paragraph)
Comment 6: For HGS measurement, specifying the posture and hand position (e.g., seated or standing) would improve reproducibility.
ANSWER: Thank you for this helpful comment. We have now added a detailed description of the participant’s posture and hand position during HGS assessment. All participants were measured in a seated position, with the elbow flexed at approximately 90 degrees, forearm and wrist in neutral positions, and feet flat on the floor. This clarification has been included to enhance the reproducibility of the methodology.
Comment 7: The study does not appear to consider variations in HGS due to measurement conditions. HGS fluctuates depending on factors such as the time of day and testing environment. For example, grip strength tends to be lower in the morning and increases throughout the day. Additionally, factors such as whether the measurement was taken after eating or while fasting, and whether participants were well-rested or fatigued, could have influenced the results. Were the measurement conditions standardized to minimize these variations?
ANSWER: We appreciate the reviewer’s attention to this important aspect. While no formal controls were applied for factors such as fasting status or prior rest, all measurements were conducted under similar conditions: during regular morning university hours (between 9:00 a.m. and 2:00 p.m.), in a quiet room, and under the supervision of trained staff. These measures aimed to minimize external variability and ensure consistency across participants. This clarification has now been included in the methodology section.
Comment 8: "The GHQ-12 is a mental health screening test." It would be beneficial to discuss whether this cutoff score is appropriate for the study population (university students) by comparing it with previous studies.
ANSWER: Thank you for this valuable suggestion. We have expanded the description of the GHQ-12 to justify the use of the ≥12 cutoff score in our university student population. This threshold has been supported in previous research involving similar samples, where it effectively identified psychological distress and emotional symptoms. A corresponding reference has been added to support this decision.
Comment 9: The analysis presented is insufficient and does not adequately explore the relationships between the variables. At a minimum, the following analyses should be included in this study:
Correlation analysis: A correlation matrix showing the relationships between hand grip strength, sleep indices, subjective perception of health, and other relevant factors.
Regression analysis: Univariate and multiple linear or logistic regression analyses should be performed, with subjective perception of health as the outcome variable. These analyses should compare the contribution of grip strength, sleep indices, and other relevant factors to the outcome. This will help determine the relative importance of each factor in predicting subjective health perception.
ANSWER: A correlation matrix between PISQ, GHQ, IPAQ and HGS of both dominant and non-dominant hands has been added (Table 6). Multivariate analysis showed that the PISQ score was the only variable that was independently associated with the GHQ-12 score. For the other associations, statistical significance was not reached. (Results, last paragraph). This affirmation has been also added to the Limitations section, as it is probably due to an insufficient small sample size (Section 4.1, 1st paragraph).
Comment 10: When explaining the relationship with subjective health perception, it is essential to consider the interaction effect between grip strength and sleep indices. Investigating whether the impact of grip strength on subjective health perception varies depending on the level of sleep quality, or vice versa, would provide a more nuanced understanding of their interplay. Such an interaction analysis is crucial for a comprehensive interpretation of the results.
ANSWER: A significant inverse correlation could be established between total PSQI score and HGS (Section 3.5). Moreover, a paragraph indicating a significant direct correlation could be established between total PISQ and GHQ-12 scores has been added (section 3.8). In fact, the association between PISQ and GHQ-12 was the only one statistically significant in the multivariate analysis (Section 3.8, last paragraph).
Results section:
Comment 11: Table 1 currently presents only the results of physical activity. To provide a more comprehensive understanding of the participants' characteristics, it would be beneficial to include data on various other factors that define the characteristics of the subjects. By including a wider range of relevant variables, the table could offer a more detailed profile of the study population
ANSWER: Sociodemographic and academic data of the sample have been added to the table with the results of physical activity.
Comment 12: Visualizing the GHQ-12 score distribution (e.g., using a histogram or box plot) would enhance clarity.
ANSWER: A histogram with the GHQ-12 score distribution has been added as Figure 1.
Comment 13: There are two sections labeled as '3.4' in the results section: '3.4. Hand grip strength assessed by dynamometer' and '3.4. Association between sleep quality and hand grip strength.' To ensure clarity and consistency, it would be advisable to renumber the second '3.4' appropriately.
ANSWER: It has been corrected.
Comment 14: Table 3 shows a statistically significant difference in HGS between males and females (p = 0.000). However, this result is biologically expected and does not provide novel insight. Since sex-based differences in muscle strength are well-established, emphasizing this finding as a key result may not add substantial value. Would it be more appropriate to shift the focus to other, more meaningful comparisons within the study?
ANSWER: We agree with the reviewer´s opinion and we have removed the table.
Comment 15: In several instances, the paper reports p-values as "p = 0.000" (e.g., in Table 3 and related text in the Results section). This is incorrect because p-values represent probabilities and can never be exactly zero. Instead, the appropriate way to report these values is "p < 0.001", which follows scientific reporting standards.
ANSWER: It has been corrected.
Comment 16: The notation of p-values in the statistical results is inconsistent in terms of uppercase and lowercase letters (e.g., 'P' vs. 'p'). Ensuring uniform formatting throughout the manuscript would improve clarity and professionalism.
ANSWER: It has been corrected.
Discussion section:
Comment 17: The association between HGS and psychological health is discussed; however, further clarification on the validity of using GHQ-12 as a psychological health indicator would strengthen the argument.
ANSWER: The following paragraph has been added “The GHQ-12 is widely recognized as a valid and reliable tool for assessing psychological health. It is designed to screen for general psychiatric morbidity, such as anxiety, depression, and social dysfunction, in both clinical and non-clinical populations. The GHQ-12 has been extensively validated across diverse populations and cultural contexts. Studies have shown it has high internal consistency (Cronbach's alpha above 0.8), indicating reliability. Furthermore, it has demonstrated good sensitivity and specificity in detecting psychological distress, making it effective for screening purposes. Its brevity (12 items) makes it practical for large-scale surveys and clinical settings, reducing respondent burden while maintaining accuracy and it has been translated into multiple languages and adapted for various cultural settings, further supporting its validity as a universal screening tool [20].” (Discussion, 3rd paragraph)
Summary:
This study provides valuable insights into the relationship between hand grip strength, sleep quality, and perceived health status in university students. However, clarity, methodological rigor, and consistency need improvement. Key revisions should focus on title specificity, statistical control of lifestyle factors, standardization of HGS measurement conditions, and proper p-value reporting. Strengthening the logical flow and scientific precision will enhance the manuscript’s impact.
Round 2
Reviewer 1 Report
Comments and Suggestions for Authors
The authors have adequately addressed all my comments/suggestions and I believe the article could be ready for publication.
Author Response
Thank you for your revision and your consideration. Your comments have been very valueable to improve the quality of the manuscript.
Reviewer 2 Report
Comments and Suggestions for Authors
Lifestyle Factors as Confounders (Comment 5)
Thank you for clearly noting the absence of data on lifestyle factors and including it in the limitations section.
To further strengthen the interpretation, a brief mention in the discussion of how such factors might potentially affect both sleep and mental health (e.g., via alcohol or caffeine) would provide helpful context, especially since these are well-established contributors.
Interaction Analysis Not Fully Addressed (Comment 10)
While the authors have added valuable correlation and multivariate analyses involving HGS, PSQI, and GHQ-12, the specific interaction effect between hand grip strength and sleep quality was not examined. For example, it remains unclear whether the association between HGS and perceived mental health varies depending on sleep quality, or vice versa.
Although such an analysis (e.g., including an interaction term HGS × PSQI in a regression model) may be limited by the current sample size, briefly acknowledging this as a potential avenue for future research would enhance the interpretive depth of the discussion.
Comment 11 (Revised)
Table 1 includes both "Males" and "Females", but since the proportion of one category logically implies the other, listing only one (e.g., "Male: n (%)") may be sufficient for simplicity.
Conclusion
Overall, your revisions have significantly improved the clarity and scientific rigor of the manuscript. The study makes a valuable contribution to understanding the complex interplay between hand grip strength, sleep quality, and psychological well-being in a university student population. Addressing the remaining points would further enhance the impact and interpretability of the findings.
Author Response
Lifestyle Factors as Confounders (Comment 5)
Thank you for clearly noting the absence of data on lifestyle factors and including it in the limitations section.
To further strengthen the interpretation, a brief mention in the discussion of how such factors might potentially affect both sleep and mental health (e.g., via alcohol or caffeine) would provide helpful context, especially since these are well-established contributors.
ANSWER: Thank you for the comment. The following paragraph has been added “Lifestyle factors such as alcohol consumption or caffeine intake, which have not been recorded in the present study, might potentially affect both sleep and mental health. Alcohol consumption can significantly impact mental health and sleep patterns. It may initially promote relaxation, but excessive intake can lead to increased anxiety and depression, as alcohol disrupts neurotransmitter balance in the brain. It also interferes with sleep quality by suppressing REM sleep, resulting in fatigue and reduced cognitive function the next day. Over time, habitual alcohol use can contribute to insomnia, creating a cycle of poor sleep and declining mental well-being [32]. On the other hand, caffeine consumption has notable effects on mental health and sleep, acting as a stimulant that temporarily boosts alertness, mood, and energy by blocking adenosine, a neurotransmitter responsible for promoting sleepiness. While moderate intake is generally considered safe and even beneficial for focus and productivity, excessive consumption can lead to heightened anxiety, restlessness, and irritability, particularly in individuals sensitive to caffeine. Regarding sleep, caffeine can disrupt sleep patterns by delaying the onset of sleep, reducing overall sleep duration, and decreasing the quality of restorative deep sleep, especially if consumed later in the day [33]. ” (Discussion, 9th paragraph)
Interaction Analysis Not Fully Addressed (Comment 10)
While the authors have added valuable correlation and multivariate analyses involving HGS, PSQI, and GHQ-12, the specific interaction effect between hand grip strength and sleep quality was not examined. For example, it remains unclear whether the association between HGS and perceived mental health varies depending on sleep quality, or vice versa.
Although such an analysis (e.g., including an interaction term HGS × PSQI in a regression model) may be limited by the current sample size, briefly acknowledging this as a potential avenue for future research would enhance the interpretive depth of the discussion.
ANSWER: Thank you for your appreciation. The following paragraph has been added as a Limitation of the study “The small sample size has also prevented for conducting a regression model to assess whether the association between HGS and perceived mental health varies depending on sleep quality, or vice versa. Future research must consider the possibility of analyzing this interaction. ” (Limitations and strengths section, 4th paragraph)
Comment 11 (Revised)
Table 1 includes both "Males" and "Females", but since the proportion of one category logically implies the other, listing only one (e.g., "Male: n (%)") may be sufficient for simplicity.
ANSWER: The data about females have been removed. (Table 1)